# Meta-learning for Positive-unlabeled Classification

## Abstract

We propose a meta-learning method for positive and unlabeled (PU) classification, which improves the performance of binary classifiers obtained from only PU data in unseen target tasks. PU learning is an important problem since PU data naturally arise in real-world applications such as outlier detection and information retrieval. Existing PU learning methods require many PU data, but sufficient data are often unavailable in practice. The proposed method minimizes the test classification risk after the model is adapted to PU data by using related tasks that consist of positive, negative, and unlabeled data. We formulate the adaptation as an estimation problem of the Bayes optimal classifier, which is an optimal classifier to minimize the classification risk. The proposed method embeds each instance into a task-specific space using neural networks. With the embedded PU data, the Bayes optimal classifier is estimated through density-ratio estimation of PU densities, whose solution is obtained as a closed-form solution. The closed-form solution enables us to efficiently and effectively minimize the test classification risk. We empirically show that the proposed method outperforms existing methods with one synthetic and three real-world datasets.

## 1 Introduction

Positive and unlabeled (PU) learning addresses a problem of learning a binary classifier from only PU data without negative data (Bekker & Davis, 2020; Elkan & Noto, 2008; Liu et al., 2003). It has attracted attention since PU data naturally arise in many real-world applications. For example, in outlier detection, only normal (positive) and unlabeled data are often available since anomalous (negative) data are difficult to collect due to their rarity. In information retrieval for images, a user selects a set of favorite images (positive data) to automatically discover relevant photos in the user's photo album (unlabeled data). In addition to these, PU learning can be applied to many other applications such as medical diagnosis (Zuluaga et al., 2011), personalized advertising (Hsieh et al., 2015), drug discovery (Liu et al., 2017), text classification (Li & Liu, 2003), and remote sensing (Li et al., 2010).

Although PU learning is attractive, it requires a large amount of PU data to learn accurate classifiers. In some real-world applications, enough data might be unavailable. For example, when we want to build personalized outlier detectors with user activity data (Hu et al., 2017; Alowais & Soon, 2012), sufficient normal data are unlikely to have been accumulated from new users. In image retrieval, users might select only a small number of favorite images. To improve performance on target tasks with limited data, meta-learning has recently been receiving a lot of attention because it learns how to learn from limited data by using labeled data on different but related tasks (source tasks) (Hospedales et al., 2021). In the above examples, for outlier detection, both normal and anomalous data might be available from other users who have existed for a long time. In image retrieval, some users might tag unfavorite photos as well as favorite ones. [1] By meta-learning, we can quickly adapt to unseen tasks (e.g., users). Although many meta-learning methods have been proposed and have performed well on various problems (Hospedales et al., 2021), they are not designed for PU learning from limited PU data.

In this paper, we propose a meta-learning method for PU classification, which meta-learns with multiple source tasks that consist of positive, negative, and unlabeled data, and uses the learned knowledge to obtain

---

[1] We discuss other potential applications of the proposed method in Section E of the appendix.

classifiers on unseen target tasks that consist of only PU data. In each meta-learning step, our model is adapted to given task-specific PU data, called a support set. Then, common parameters shared across all tasks are meta-learned to minimize the test classification risk (the expected test misclassification rate) when the adapted model is used. This training procedure enables our model to learn how to learn from PU data.

More specifically, the adaptation to the support set corresponds to estimating a *Bayes optimal classifier*, which is an optimal classifier to minimize the classification risk. To estimate this with the support set, we use the fact that the Bayes optimal classifier can be represented by two components: a density-ratio between PU densities and a positive class-prior. With the proposed method, the density-ratio is modeled by using neural networks that have high expressive capabilities. To effectively adapt to a wide variety of tasks, task-specific density-ratio models need to be constructed. To this end, the proposed method first calculates task representation vectors of the support set by permutation-invariant neural networks that can take a set of instances as input (Zaheer et al., 2017). With the task representations, each instance can be non-linearly embedded into a task-specific space by other neural networks. The proposed method performs density-ratio estimation by using the embedded support set, and its solution is obtained as a closed-form solution. By using the estimated density-ratio, the positive class-prior can also be quickly estimated. As a result, the proposed method can efficiently and effectively estimate the Bayes optimal classifier by using only the closed-form solution of the density-ratio estimation. This efficient and effective adaptation is a major strength of our proposed model since meta-learning usually takes high computation costs (Bertinetto et al., 2018; Lee et al., 2019b).

We meta-learn all the neural networks such that the test classification risk, which is directly calculated with positive and negative data, is minimized when the estimated Bayes optimal classifier is used. All parameters of the neural networks are common parameters shared among all tasks. Since the closed-form solution of the density-ratio estimation is differentiable, we can solve the whole optimization problem by a stochastic gradient descent method. By minimizing the classification risk on various tasks, we can learn accurate classifiers for unseen target tasks from only target PU data.

Our main contributions are summarized as follows:

- To the best of our knowledge, our work is the first attempt at meta-learning for positive-unlabeled classification with a few PU data.

- We propose an efficient and effective meta-learning method that estimates the task-specific Bayes optimal classifier using only the closed-form solution of the density-ratio estimation.

- We empirically show that the proposed method outperformed existing PU learning methods and their meta-learning variants when there is insufficient PU data in both synthetic and real-world datasets.

## 2 Related Work

Many PU learning methods have been proposed (Bekker & Davis, 2020; Fung et al., 2005; Hou et al., 2018; Chen et al., 2020a; Luo et al., 2021; Zhao et al., 2022; Wang et al., 2023). Early methods used some heuristics to identify negative instances in unlabeled data (Li & Liu, 2003; Liu et al., 2003; Yu et al., 2002). However, the performances of these methods heavily rely on the heuristic strategy and data separability assumption, i.e., positive and negative distributions are separated (Nakajima & Sugiyama, 2023). Recently, state-of-the art performances have been achieved by empirical risk minimization-based methods, which rewrite the classification risk with PU data and learn classifiers by minimizing the risk (Du Plessis et al., 2015; Kiryo et al., 2017; Du Plessis et al., 2014). Unlike the heuristic approach, they can build statistically-consistent classifiers (Du Plessis et al., 2015; Kiryo et al., 2017; Du Plessis et al., 2014). A few methods use density-ratio estimation for obtaining the Bayes optimal classifier, which minimizes the classification risk, and have shown excellent performance (Charoenphakdee & Sugiyama, 2019; Nakajima & Sugiyama, 2023). The proposed method also uses the density-ratio estimation-based approach for the support set adaptation since it enables us to obtain the closed-form solution, which is vital for efficient and effective meta-learning as described

later. Although the empirical risk minimization-based methods are useful, they require the class-prior probability information. In real-world applications, it is difficult to know the class-prior without negative data in advance. Thus, the class-prior needs to be estimated from only PU data. However, the class-prior estimation with PU data is known as an ill-posed problem (Blanchard et al., 2010; Scott, 2015). To overcome this, *the irreducible assumption* is commonly used, which states the support of the positive-conditional distribution is not contained in the support of the negative-conditional distribution (Yao et al., 2021; Ivanov, 2020; Ramaswamy et al., 2016; Blanchard et al., 2010; Scott, 2015). The proposed method also uses this assumption and estimates the class-prior on the basis of the estimated density-ratio. Unlike the proposed method, all these PU learning methods are designed for one task and cannot use data in related tasks.

PNU learning uses positive, negative, and unlabeled (PNU) data in a task to learn binary classifiers (Sakai et al., 2017; Hsieh et al., 2019; Sakai et al., 2018). Especially, (Sakai et al., 2017) uses the technique of PU learning to rewrite the classification risk with PNU data, which enables us to learn binary classifiers without particular distributional assumptions such as the cluster assumption (Grandvalet & Bengio, 2004). Unlike the proposed method, these methods cannot use data in source tasks and require negative data in the target tasks.

Some studies use PU learning for domain adaptation, which learns classifiers that fit on a target task by using data in a source task (Sonntag et al., 2022; Loghmani et al., 2020; Sakai & Shimizu, 2019; Hammoudeh & Lowd, 2020). These all methods treat only two tasks (source and target tasks) and require both target and source data in the training phase. In contrast, the proposed method can treat multiple tasks and does not require any target data in the (meta-)training phase, which is preferable when new target tasks appear one after another. In addition, these methods are designed for the setting where class labels are the same in source and target tasks (domains) although the proposed method can treat tasks that have different class labels.

Meta-learning methods train a model such that it generalizes well after adapting to a few data by using data on multiple tasks (Finn et al., 2017; Snell et al., 2017; Garnelo et al., 2018; Rajeswaran et al., 2019; Iwata & Kumagai, 2020; Jiang et al., 2022). Although many meta-learning methods have been proposed, to our knowledge, no meta-leaning methods have been designed for PU learning from a few PU data. Although one method in (Chen et al., 2020b) uses a technique of the meta-learning for weighting unlabeled data in PU learning, it cannot treat source tasks and is not designed for learning from a few PU data. A representative method for meta-learning is model-agnostic meta-learning (MAML) (Finn et al., 2017), which adapts to support data by using an iterative gradient method. It requires second-order derivatives of the parameters of whole neural networks and needs to retain all computation graphs of the iterative adaptation during meta-training, which imposes considerable computational and memory burdens (Bertinetto et al., 2018). The proposed method is more efficient than MAML since it adapts to support data by using only a closed-form solution of the density-ratio estimation. Although some methods formulate the adaptation as the convex optimization problem for efficient meta-learning, almost all methods consider ordinary classification tasks (Bertinetto et al., 2018; Lee et al., 2019b). A meta-learning method for the relative density-ratio estimation has been proposed (Kumagai et al., 2021). Although this method estimates relative density-ratio from a few data by using multiple unlabeled datasets, it cannot estimate classifiers from PU data. In contrast, the proposed method can estimate the classifier by sequentially estimating the density-ratio of PU data and the positive class-prior in the support set adaptation.

## 3 Preliminary

We briefly explain the concept of PU learning based on density-ratio estimation. Let $X \in \mathbb{R}^D$ and $Y \in \{\pm 1\}$ be the input and output random variables, where $D$ is the dimensionality of the input variable. Let $p(\mathbf{x}, y)$ be the joint density of $(X, Y)$, $p^{\mathrm{p}}(\mathbf{x}) = p(\mathbf{x}|Y = +1)$ and $p^{\mathrm{n}}(\mathbf{x}) = p(\mathbf{x}|Y = -1)$ be the positive and negative class-conditional densities, $p(\mathbf{x}) = \pi^{\mathrm{p}}p^{\mathrm{p}}(\mathbf{x}) + (1-\pi^{\mathrm{p}})p^{\mathrm{n}}(\mathbf{x})$ be the input marginal density, and $\pi^{\mathrm{p}} = p(Y = 1)$ be the positive class-prior probability. $s : \mathbb{R}^D \to \{\pm 1\}$ is a binary classifier and $l_{01} : \mathbb{R} \times \{\pm 1\} \to \mathbb{R}$ is the zero-one loss, $l_{01}(t, y) = (1 - \mathrm{sign}(ty))/2$, where $\mathrm{sign}(\cdot)$ is a sign function; $\mathrm{sign}(U) = 1$ if $U \geq 0$ and $\mathrm{sign}(U) = -1$ otherwise. PU learning aims to learn binary classifier $s$ from only PU data, which minimizes

the classification risk:

$$R_{\mathrm{pn}}(s) = \mathbb{E}_{(X,Y)\sim p(\mathbf{x},y)}[l_{01}(s(X),Y)] = \pi^{\mathrm{p}}\mathbb{E}_{X\sim p^{\mathrm{p}}}[l_{01}(s(X),+1)] + (1-\pi^{\mathrm{p}})\mathbb{E}_{X\sim p^{\mathrm{n}}}[l_{01}(s(X),-1)], \qquad (1)$$

where $\mathbb{E}$ denotes an expectation. It is known that the optimal solution for $R_{\mathrm{pn}}(s)$ can be written as

$$s_*(\mathbf{x}) = \mathrm{sign}(p(Y=1|\mathbf{x})-0.5) = \mathrm{sign}\left(\pi^{\mathrm{p}}\frac{p^{\mathrm{p}}(\mathbf{x})}{p(\mathbf{x})}-0.5\right), \qquad (2)$$

where Bayes' theorem is used in the second equality (Charoenphakdee & Sugiyama, 2019; Nakajima & Sugiyama, 2023). This solution is called the Bayes optimal classifier. Since density-ratio $r(\mathbf{x}) = \frac{p^{\mathrm{p}}(\mathbf{x})}{p(\mathbf{x})}$ depends on only PU densities, we can estimate $s_*(\mathbf{x})$ with only PU data through the density-ratio estimation when class-prior $\pi^{\mathrm{p}}$ is known.

## 4 Proposed Method

In this section, we explain the proposed method. This section is organized as follows: In Section 4.1, we present our problem formulation. In Section 4.2, we present our model to estimate the Bayes optimal classifier given PU data. Our model consists of task representation calculation, density-ratio estimation, and class-prior estimation. In Section 4.3, we explain our meta-learning procedure to train our model. Figure 1 shows the overview of our meta-learning procedure.

### 4.1 Problem Formulation

Let $\mathcal{D}_t^{\mathrm{p}} := \{\mathbf{x}_{tn}^{\mathrm{p}}\}_{n=1}^{N_t^{\mathrm{p}}} \sim p_t^{\mathrm{p}}$ be a set of positive instances in the $t$-th task, where $\mathbf{x}_{tn}^{\mathrm{p}} \in \mathbb{R}^D$ is the $D$-dimensional feature vector of the $n$-th positive instance of the $t$-th task, $N_t^{\mathrm{p}}$ is the number of the positive instances in the $t$-th task, and $p_t^{\mathrm{p}}$ is the positive density of the $t$-th task. Similarly, let $\mathcal{D}_t^{\mathrm{n}} := \{\mathbf{x}_{tn}^{\mathrm{n}}\}_{n=1}^{N_t^{\mathrm{n}}} \sim p_t^{\mathrm{n}}$ and $\mathcal{D}_t^{\mathrm{u}} := \{\mathbf{x}_{tn}^{\mathrm{u}}\}_{n=1}^{N_t^{\mathrm{u}}} \sim p_t = \pi_t^{\mathrm{p}}p_t^{\mathrm{p}} + (1-\pi_t^{\mathrm{p}})p_t^{\mathrm{n}}$ be negative and unlabeled instances of the $t$-th task, respectively, where $\pi_t^{\mathrm{p}}$ is a positive class-prior of the $t$-th task. In the meta-training phase, we are given $T$ source tasks $\mathcal{D} := \{\mathcal{D}_t^{\mathrm{p}} \cup \mathcal{D}_t^{\mathrm{n}} \cup \mathcal{D}_t^{\mathrm{u}}\}_{t=1}^{T}$. We assume that feature vector size $D$ is the same across all tasks, and all tasks are drawn from the same task distribution although joint distribution $p_t(\mathbf{x},y)$ can vary across tasks. These assumptions are common in meta-learning studies (Finn et al., 2017; Snell et al., 2017; Garnelo et al., 2018; Rajeswaran et al., 2019; Kumagai et al., 2023; Farid & Majumdar, 2021). Since $p_t(\mathbf{x},y)$ can differ across tasks, we cannot directly use labeled data in source tasks to learn classifiers in target tasks. In the test phase, we are given PU data (support set) in a target task $\mathcal{S} = \mathcal{S}^{\mathrm{p}} \cup \mathcal{S}^{\mathrm{u}} = \{\mathbf{x}_n^{\mathrm{p}}\}_{n=1}^{N^{\mathrm{p}}} \cup \{\mathbf{x}_n^{\mathrm{u}}\}_{n=1}^{N^{\mathrm{u}}}$, which is different from source tasks. Our aim is to learn a binary classifier that minimizes the test classification risk on the target task to accurately classify any test instance $\mathbf{x}$ drawn from the same distribution as unlabeled support data $\mathcal{S}^{\mathrm{u}}$.

### 4.2 Model

In this section, we explain how to estimate the task-specific Bayes optimal classifier given support set $\mathcal{S}$. We omit task index $t$ for simplicity since all procedures are conducted in a task in this section.

#### 4.2.1 Task Representation Calculation

First, we explain how to obtain a vector representation of the given dataset, which is used for obtaining task-specific instance embeddings appropriate for the task. Specifically, the proposed method calculates $M$-dimensional task representation vectors, $\mathbf{z}^{\mathrm{p}}$ and $\mathbf{z}^{\mathrm{u}}$, of the support set $\mathcal{S}$ using the following permutation-invariant neural networks (Zaheer et al., 2017):

$$\mathbf{z}^{\mathrm{p}} = g\left(\frac{1}{N^{\mathrm{p}}}\sum_{n=1}^{N^{\mathrm{p}}} f(\mathbf{x}_n^{\mathrm{p}})\right), \quad \mathbf{z}^{\mathrm{u}} = g\left(\frac{1}{N^{\mathrm{u}}}\sum_{n=1}^{N^{\mathrm{u}}} f(\mathbf{x}_n^{\mathrm{u}})\right), \qquad (3)$$

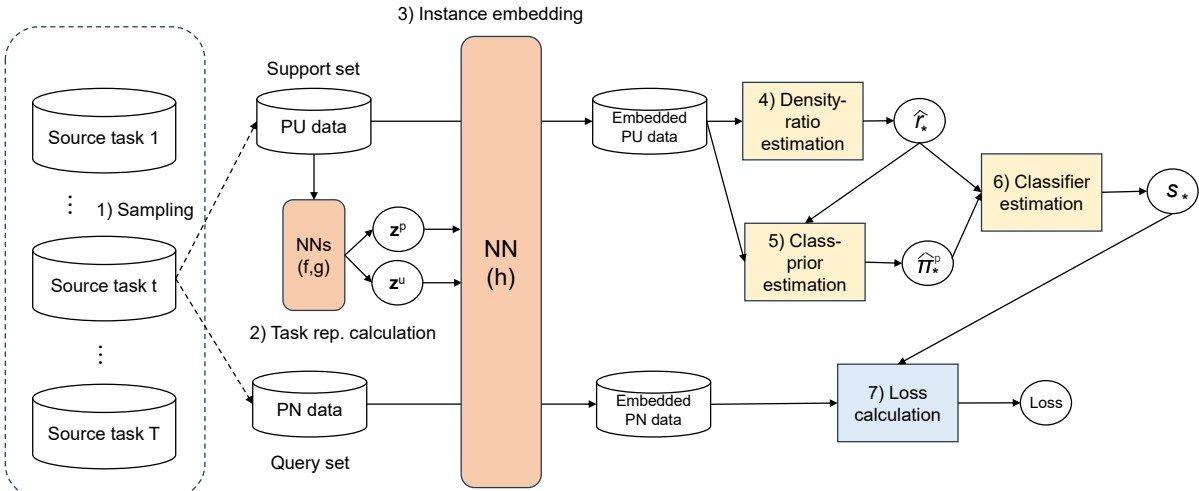

Figure 1: Our meta-learning procedure: (1) For each training iteration, we randomly sample PU data (support set) and positive and negative (PN) data (query set) from a randomly selected source task. (2) Task representation vectors, $\mathbf{z}^{\mathrm{p}}$ and $\mathbf{z}^{\mathrm{u}}$, are calculated from positive and unlabeled support data, respectively, by using the permutation-invariant neural networks (Section 4.2.1). (3) With $\mathbf{z}^{\mathrm{p}}$ and $\mathbf{z}^{\mathrm{u}}$, all instances are embedded into a task-specific space by using a neural network. (4) By using the embedded support set, density-ratio estimation is performed and its closed-form solution $\hat{r}_*$ is obtained (Section 4.2.2). (5) By using $\hat{r}_*$ and the embedded support set, positive class-prior $\hat{\pi}_*^{\mathrm{p}}$ is estimated (Section 4.2.3). (6) By using $\hat{r}_*$ and $\hat{\pi}_*^{\mathrm{p}}$, the estimated Bayes optimal classifier $s_*$ is obtained (Section 4.2.3). (7) Test classification risk (loss) is calculated with the query set and the estimated Bayes optimal classifier, and it can be backpropagated to update all the neural networks (Section 4.3).

where $f$ and $g$ are any feed-forward neural network. Since summation is permutation-invariant, the neural network in Eq. (3) outputs the same vector even when the order of instances varies. In addition, this neural network can handle different numbers of instances. Thus, this neural network can be used as functions for set inputs. Task representation vectors $\mathbf{z}^{\mathrm{p}}$ and $\mathbf{z}^{\mathrm{u}}$ contain information of the PU data. The proposed method uses the task representation vectors to obtain task-specific instance embeddings to deal with the diversity of tasks. The proposed method can use any other permutation-invariant function such as summation (Zaheer et al., 2017) and set transformer (Lee et al., 2019a) to obtain task representation vectors.

### 4.2.2 Density-ratio Estimation

Next, we explain how to estimate density-ratio $r(\mathbf{x}) = \frac{p^{\mathrm{p}}(\mathbf{x})}{p(\mathbf{x})}$ with $\mathbf{z}^{\mathrm{p}}$ and $\mathbf{z}^{\mathrm{u}}$. A naive approach is to estimate each density and then take the ratio. However, it does not work well since density estimation is known as a difficult problem and taking the ratio of the two estimations can increase the error (Vapnik, 1999; Sugiyama et al., 2012). Instead, direct density-ratio estimation without going through density estimation has become the standard approach (Kanamori et al., 2009; Sugiyama et al., 2012; Yamada et al., 2013). In particular, neural networks have been recently used for modeling the density-ratio due to their high flexibility and expressibility (Rhodes et al., 2020; Kato & Teshima, 2021). Following this, our method directly models the density-ratio $r(\mathbf{x}) = \frac{p^{\mathrm{p}}(\mathbf{x})}{p(\mathbf{x})}$ by the following neural network,

$$\hat{r}(\mathbf{x}; \mathcal{S}) = \mathbf{w}^{\top} h([\mathbf{x}, \mathbf{z}^{\mathrm{p}}, \mathbf{z}^{\mathrm{u}}]), \tag{4}$$

where $[\cdot, \cdot, \cdot]$ is a concatenation, $h : \mathbb{R}^{D+2K} \to \mathbb{R}_{>0}^M$ is a feed-forward neural network for instance embedding, and $\mathbf{w} \in \mathbb{R}_{\geq 0}^M$ is linear weights. Since both the outputs of $h$ and $\mathbf{w}$ are non-negative, the estimated density-ratio is also non-negative. $h([\mathbf{x}, \mathbf{z}^{\mathrm{p}}, \mathbf{z}^{\mathrm{u}}])$ depends on both $\mathbf{z}^{\mathrm{p}}$ and $\mathbf{z}^{\mathrm{u}}$, and thus, $h([\mathbf{x}, \mathbf{z}^{\mathrm{p}}, \mathbf{z}^{\mathrm{u}}])$ represents the task-specific embedding of instance $\mathbf{x}$. Liner weights $\mathbf{w}$ and parameters of all neural networks $f$, $g$, and $h$ are task-specific and task-shared parameters, respectively. Task-shared parameters are meta-learned to

improve the expected test performance, which is explained in Section 4.3. Only task-specific parameters $\mathbf{w}$ are adapted to given support set $\mathcal{S}$, which enables us to estimate the density-ratio as a closed-form solution.

Linear weights $\mathbf{w}$ are determined so that the following expected squared error between the true density-ratio $r(\mathbf{x})$ and estimated density-ratio $\hat{r}(\mathbf{x}; \mathcal{S})$ is minimized:

$$J(\mathbf{w}) = \frac{1}{2}\mathbb{E}_{X \sim p}\left[(r(X) - \hat{r}(X; \mathcal{S}))^2\right] = \frac{1}{2}\mathbb{E}_{X \sim p}\left[\hat{r}(X; \mathcal{S})^2\right] - \mathbb{E}_{X \sim p^{\mathrm{p}}}\left[\hat{r}(X; \mathcal{S})\right] + \mathrm{Const.}, \tag{5}$$

where Const. is a constant term that does not depend on our model. By approximating the expectation with support set $\mathcal{S}$ and excluding the non-negative constraints of $\mathbf{w}$, we obtain the following optimization problem:

$$\tilde{\mathbf{w}} = \arg\min_{\mathbf{w} \in \mathbb{R}^M}\left[\frac{1}{2}\mathbf{w}^\top \mathbf{K}\mathbf{w} - \mathbf{k}^\top \mathbf{w} + \frac{\lambda}{2}\mathbf{w}^\top \mathbf{w}\right], \tag{6}$$

where $\mathbf{k} = \frac{1}{N^{\mathrm{p}}}\sum_{n=1}^{N^{\mathrm{p}}} h([\mathbf{x}_n^{\mathrm{p}}, \mathbf{z}^{\mathrm{p}}, \mathbf{z}^{\mathrm{u}}])$, $\mathbf{K} = \frac{1}{N^{\mathrm{u}}}\sum_{n=1}^{N^{\mathrm{u}}} h([\mathbf{x}_n^{\mathrm{u}}, \mathbf{z}^{\mathrm{p}}, \mathbf{z}^{\mathrm{u}}])h([\mathbf{x}_n^{\mathrm{u}}, \mathbf{z}^{\mathrm{p}}, \mathbf{z}^{\mathrm{u}}])^\top$, the third term of r.h.s. of Eq. (6) is the $\ell^2$-regularizer to prevent over-fitting, and $\lambda > 0$ is a positive constant. The global optimum solution for Eq. (6) can be obtained as the following closed-form solution:

$$\tilde{\mathbf{w}} = (\mathbf{K} + \lambda\mathbf{I})^{-1}\mathbf{k}, \tag{7}$$

where $\mathbf{I}$ is the $M$ dimensional identity matrix. This closed-form solution can be efficiently obtained when $M$ is not large. We note that $(\mathbf{K} + \lambda\mathbf{I})^{-1}$ exists since $\lambda > 0$ makes $(\mathbf{K} + \lambda\mathbf{I})$ positive-definite. Since we omitted the non-negative constraints of $\mathbf{w}$ in Eq. (6), some learned weights $\tilde{\mathbf{w}}$ can be negative. To compensate for this, following previous studies (Kanamori et al., 2009), the solution is modified as $\hat{\mathbf{w}} = \max(\mathbf{0}, \tilde{\mathbf{w}})$, where max operator is applied in an element-wise manner. By using the learned weights, the density-ratio estimated with support set $\mathcal{S}$ is obtained as

$$\hat{r}_*(\mathbf{x}; \mathcal{S}) = \hat{\mathbf{w}}^\top h([\mathbf{x}, \mathbf{z}^{\mathrm{p}}, \mathbf{z}^{\mathrm{u}}]). \tag{8}$$

### 4.2.3 Class-prior Estimation

We explain how to estimate class-prior $\pi^{\mathrm{p}}$ from $\mathcal{S}$ by using estimated density-ratio $\hat{r}_*(\mathbf{x}; \mathcal{S})$. To view the relationship between the density-ratio and the class-prior, we first consider the following equation as in (Blanchard et al., 2010; Scott, 2015):

$$\frac{1}{r(\mathbf{x})} = \frac{p(\mathbf{x})}{p^{\mathrm{p}}(\mathbf{x})} = \frac{\pi^{\mathrm{p}}p^{\mathrm{p}}(\mathbf{x}) + (1 - \pi^{\mathrm{p}})p^{\mathrm{n}}(\mathbf{x})}{p^{\mathrm{p}}(\mathbf{x})} = \pi^{\mathrm{p}} + (1 - \pi^{\mathrm{p}})\frac{p^{\mathrm{n}}(\mathbf{x})}{p^{\mathrm{p}}(\mathbf{x})}, \tag{9}$$

where $\mathbf{x} \in \{\mathbf{x}'|p^{\mathrm{p}}(\mathbf{x}') > 0\}$. Since $\frac{p^{\mathrm{n}}(\mathbf{x})}{p^{\mathrm{p}}(\mathbf{x})}$ is non-negative, this equation implies

$$\inf_{\mathbf{x};p^{\mathrm{p}}(\mathbf{x})>0}\frac{1}{r(\mathbf{x})} \geq \pi^{\mathrm{p}}. \tag{10}$$

When we can assume that the support of $p^{\mathrm{p}}$ is not contained in the support of $p^{\mathrm{n}}$, i.e., $\{\mathbf{x}|(p^{\mathrm{p}}(\mathbf{x}) > 0) \wedge (p^{\mathrm{n}}(\mathbf{x}) = 0)\} \neq \emptyset$, the above equality holds; $\inf_{\mathbf{x};p^{\mathrm{p}}(\mathbf{x})>0}\frac{1}{r(\mathbf{x})} = \pi^{\mathrm{p}}$. This assumption is called the irreducible assumption and is widely used in PU learning studies (Yao et al., 2021; Ivanov, 2020; Ramaswamy et al., 2016; Blanchard et al., 2010; Scott, 2015). By using Eqs. (10) and (8), our method estimates class-prior $\pi^{\mathrm{p}}$ with support set $\mathcal{S}$ as

$$\tilde{\pi}_*^{\mathrm{p}}(\mathcal{S}) = \min_{\mathbf{x} \in \mathcal{S}}\frac{1}{\hat{r}_*(\mathbf{x}; \mathcal{S})} = \frac{1}{\max_{\mathbf{x} \in \mathcal{S}}\hat{r}_*(\mathbf{x}; \mathcal{S})}. \tag{11}$$

We used the max operator for $\mathcal{S}$ instead of $\mathcal{S}^{\mathrm{p}}$. This is because $\mathcal{S}$ contains instances drawn from $p^{\mathrm{p}}$ and even if $p^{\mathrm{p}}(\mathbf{x}') = 0$, $\mathbf{x}'$ does not give the maximum value due to $r(\mathbf{x}') = 0$. We note that the max operator is often used in neural networks such as max pooling (Nagi et al., 2011). Further, since $\pi^{\mathrm{p}} \leq 1$, the estimated

prior is modified as $\hat{\pi}_*^{\mathrm{p}}(\mathcal{S}) = \min(\tilde{\pi}_*^{\mathrm{p}}(\mathcal{S}), 1)$. As a result, by using Eqs. (2), (8), and (11), the Bayes optimal classifier estimated from support set $\mathcal{S}$ is obtained as

$$s_*(\mathbf{x}; \mathcal{S}) = \mathrm{sign}\left(\hat{\pi}_*^{\mathrm{p}}(\mathcal{S})\hat{r}_*(\mathbf{x}; \mathcal{S}) - 0.5\right). \tag{12}$$

Our formulation is based on only density-ratio estimation, whose solution is easily obtained as the closed-form solution. Thus, it can perform fast and effective adaptation. We note that estimating the Bayes optimal classifier, including the task representation calculation, the density-ratio estimation, and the class-prior estimation, requires only PU data. Therefore, our model can be applied to target tasks that have only PU data.

## 4.3 Meta-training

We explain the training procedure for our model. In this subsection, we use notation $\mathcal{S}$ as a support set in source tasks. The common parameters to be meta-learned $\Theta$ are parameters of neural networks $f$, $g$, and $h$, and regularization parameters $\lambda$. In the meta-learning, we want to minimize the test classification risk when the Bayes optimal classifier adapted to support set $\mathcal{S}$ is used:

$$\min_{\Theta} \ \mathbb{E}_{t \sim \{1, \ldots, T\}} \mathbb{E}_{(\mathcal{S}, \mathcal{Q}) \sim \mathcal{D}_t} \left[ R_{\mathrm{pn}}(\Theta, \mathcal{Q}; \mathcal{S}) \right], \tag{13}$$

where $\mathcal{Q} = \mathcal{Q}^{\mathrm{p}} \cup \mathcal{Q}^{\mathrm{n}}$ is a query set, where $\mathcal{Q}^{\mathrm{p}}$ is a set of positive instances and $\mathcal{Q}^{\mathrm{n}}$ is a set of negative instances drawn from the same task as support set $\mathcal{S}$, and

$$R_{\mathrm{pn}}(\Theta, \mathcal{Q}; \mathcal{S}) = \frac{\pi_{\mathcal{Q}}^{\mathrm{p}}}{|\mathcal{Q}^{\mathrm{p}}|} \sum_{\mathbf{x} \in \mathcal{Q}^{\mathrm{p}}} l_{01}(s_*(\mathbf{x}; \mathcal{S}, \Theta), +1) + \frac{1 - \pi_{\mathcal{Q}}^{\mathrm{p}}}{|\mathcal{Q}^{\mathrm{n}}|} \sum_{\mathbf{x} \in \mathcal{Q}^{\mathrm{n}}} l_{01}(s_*(\mathbf{x}; \mathcal{S}, \Theta), -1), \tag{14}$$

where $\pi_{\mathcal{Q}}^{\mathrm{p}} = \frac{|\mathcal{Q}^{\mathrm{p}}|}{|\mathcal{Q}^{\mathrm{p}}| + |\mathcal{Q}^{\mathrm{n}}|}$, and $s_*(\mathbf{x}; \mathcal{S}, \Theta)$ is the Bayes optimal classifier estimated with support set $\mathcal{S}$ in Eq. (12). Here, we explicitly describe the dependency of common parameter $\Theta$ for clarity. To simulate test environments, we sample query set $\mathcal{Q}$ so that the ratio of positive instances in the query set matches that of the original dataset, i.e., $\pi_{\mathcal{Q}}^{\mathrm{p}} = \frac{|\mathcal{Q}^{\mathrm{p}}|}{|\mathcal{Q}^{\mathrm{p}}| + |\mathcal{Q}^{\mathrm{n}}|} = \frac{|\mathcal{D}_t^{\mathrm{p}}|}{|\mathcal{D}_t^{\mathrm{p}}| + |\mathcal{D}_t^{\mathrm{n}}|}$. Since the gradient of the zero-one loss is zero everywhere except for the origin, gradient descent methods cannot be used. To avoid this, surrogate losses of the zero-one loss (e.g., the sigmoid function) are usually used (Kiryo et al., 2017; Du Plessis et al., 2015; Nakajima & Sugiyama, 2023). Following this, we use the following smoothed risk for training:

$$\widetilde{R_{\mathrm{pn}}}(\Theta, \mathcal{Q}; \mathcal{S}) = \frac{\pi_{\mathcal{Q}}^{\mathrm{p}}}{|\mathcal{Q}^{\mathrm{p}}|} \sum_{\mathbf{x} \in \mathcal{Q}^{\mathrm{p}}} \sigma_\tau(-u_*(\mathbf{x}; \mathcal{S}, \Theta)) + \frac{1 - \pi_{\mathcal{Q}}^{\mathrm{p}}}{|\mathcal{Q}^{\mathrm{n}}|} \sum_{\mathbf{x} \in \mathcal{Q}^{\mathrm{n}}} \sigma_\tau(u_*(\mathbf{x}; \mathcal{S}, \Theta)), \tag{15}$$

where $u_*(\mathbf{x}; \mathcal{S}, \Theta) = \hat{\pi}_{\mathrm{p}}^*(\mathcal{S}) \cdot \hat{r}^*(\mathbf{x}; \mathcal{S}) - 0.5$ and $\sigma_\tau(U) = \frac{1}{1 + \exp(-\tau \cdot U)}$ is the sigmoid function with scaling parameter $\tau > 0$. We can accurately approximate the zero-one loss as $\tau$ increases. In our experiments, we set $\tau = 10$. Since $u_*(\mathbf{x}; \mathcal{S}, \Theta)$ is easily obtained on the basis of the closed-form solution of the density-ratio, the risk in Eq. (13) is efficiently calculated. In addition, the risk is differentiable since $u_*(\mathbf{x}; \mathcal{S}, \Theta)$ is differentiable. Thus, we can solve it by a stochastic gradient descent method. Algorithm 1 shows the pseudocode for our training procedure. For each iteration, we randomly sample task $t$ from source tasks (Line 2). From positive data $\mathcal{D}_t^{\mathrm{p}}$ and unlabeled data $\mathcal{D}_t^{\mathrm{u}}$, we randomly sample support set $\mathcal{S} = \mathcal{S}^{\mathrm{p}} \cup \mathcal{S}^{\mathrm{u}}$ (Lines $3 - 4$). We note that even when there are no unlabeled data in a task, we can sample unlabeled data from labeled data by hiding label information. From labeled data $(\mathcal{D}_t^{\mathrm{p}} \setminus \mathcal{S}^{\mathrm{p}}) \cup \mathcal{D}_t^{\mathrm{n}}$, we randomly sample query set $\mathcal{Q}$ (Line 5) while maintaining the positive ratio $\frac{|\mathcal{D}_t^{\mathrm{p}}|}{|\mathcal{D}_t^{\mathrm{p}}| + |\mathcal{D}_t^{\mathrm{n}}|}$. We calculate task representations $\mathbf{z}^{\mathrm{p}}$ and $\mathbf{z}^{\mathrm{u}}$ from $\mathcal{S}$ (Line 6). We estimate density-ratio $r(\mathbf{x})$ and positive class-prior $\pi^{\mathrm{p}}$ from support set $\mathcal{S}$ to obtain the Bayes optimal classifier (Lines $7 - 8$). By using the estimated Bayes optimal classifier, we calculate the test classification risk on query set $\mathcal{Q}$, $\widetilde{R_{\mathrm{pn}}}(\Theta, \mathcal{Q}; \mathcal{S})$ (Line 9). Lastly, the common parameters of our model $\Theta$ are updated with the gradient of $\widetilde{R_{\mathrm{pn}}}(\Theta, \mathcal{Q}; \mathcal{S})$ (Line 10). After meta-learning, given a few PU data $\mathcal{S}$ in a target task, we can obtain the target task-specific Bayes optimal classifier by running lines 6 to 8 in Algorithm 1 with $\mathcal{S}$. Since our model is meta-learned to accurately estimate Bayes optimal classifiers from a few PU data on multiple source tasks, we can expect it to generalize for the target task.

---

**Algorithm 1** Training procedure of our model.

---

**Require:** Datasets in source tasks $\mathcal{D}$, positive support set size $N_{\mathcal{S}}^{\mathrm{p}}$, unlabeled support set size $N_{\mathcal{S}}^{\mathrm{u}}$, and query set size $N_{\mathcal{Q}}$
**Ensure:** Common parameters of our model $\Theta$
 1: **repeat**
 2:     Randomly sample task $t$ from $\{1, \ldots, T\}$
 3:     Randomly sample positive support set $\mathcal{S}^{\mathrm{p}}$ with size $N_{\mathcal{S}}^{\mathrm{p}}$ from $\mathcal{D}_t^{\mathrm{p}}$
 4:     Randomly sample unlabeled support set $\mathcal{S}^{\mathrm{u}}$ with size $N_{\mathcal{S}}^{\mathrm{u}}$ from $\mathcal{D}_t^{\mathrm{u}}$
 5:     Randomly sample query set $\mathcal{Q}$ with size $N_{\mathcal{Q}}$ from $(\mathcal{D}_t^{\mathrm{p}} \setminus \mathcal{S}^{\mathrm{p}}) \cup \mathcal{D}_t^{\mathrm{n}}$ while maintaining the positive ratio of task $t$
 6:     Calculate task representations $\mathbf{z}^{\mathrm{p}}$ and $\mathbf{z}^{\mathrm{u}}$ from $\mathcal{S}^{\mathrm{p}}$ and $\mathcal{S}^{\mathrm{u}}$, respectively by Eq. (3)
 7:     Calculate linear weights $\tilde{\mathbf{w}}$ from $\mathcal{S}$ by (7) to obtain Eq. (8)
 8:     Calculate positive class-prior $\tilde{\pi}^{\mathrm{p}}$ from $\mathcal{S}$ by (11)
 9:     Calculate the empirical test classification risk $\widetilde{R_{\mathrm{pn}}}(\Theta, \mathcal{Q}; \mathcal{S})$ by Eq. (15)
10:     Update common parameters $\Theta$ with the gradients of $\widetilde{R_{\mathrm{pn}}}(\Theta, \mathcal{Q}; \mathcal{S})$
11: **until** End condition is satisfied;

---

## 5 Experiments

### 5.1 Data

We evaluated the proposed method with one synthetic and three real-world datasets: Mnist-r, Isolet, and IoT. We used these real-world datasets since they have been commonly used in meta-learning and domain adaptation studies (Kumagai et al., 2019; 2021; 2023; Ghifary et al., 2015; Mahajan et al., 2021). The synthetic dataset has 140 tasks, where each task was created on the basis of Gaussian mixture and half moon data (Wiebe et al., 2015). Gaussian mixture consists of data drawn from a two-dimensional Gaussian mixture model, in which positive and negative distributions were $\mathcal{N}(\mathbf{x}|(-1.5, 0), \mathbf{I}_2)$, and $\mathcal{N}(\mathbf{x}|(1.5, 0), \mathbf{I}_2)$, respectively. Half moon consists of two crescent-shaped clusters of data with Gaussian noises with 0.4 variance. Each cluster corresponds to a positive or negative class. To create the $t$-th task, we generated data from a randomly selected Gaussian mixture or half moon and then rotated the generated data by $2\pi \frac{t-1}{180}$ rad around the origin. The class-prior of each task is described later. We randomly selected 100 source, 20 validation, and 20 target tasks. Mnist-r is created from Mnist dataset by rotating the images (Ghifary et al., 2015). This dataset has six domains (six rotating angles) with 10 class (digit) labels. Each class of each domain has 100 instances and its feature dimension is 256. Isolet consists of 26 letters (classes) spoken by 150 speakers, and speakers are grouped into five groups (domains) by speaking similarity (Fanty & Cole, 1990). Each instance is represented as a 617-dimensional vector. IoT is real network traffic data for outlier detection, which are generated from nine IoT devices (domains) infected by malware (Meidan et al., 2018). Each instance is represented by a 115-dimensional vector. For each domain, we randomly used 2000 normal and 2000 malicious instances. We normalized each instance by $\ell_2$ normalization.

We explain how to create binary classification tasks from each real-world dataset. We used the task construction procedure similar to the previous meta-learning study (Kumagai et al., 2023). Specifically, for Mnist-r, we first randomly split all six domains into three groups: four, one, and one domains. Then, we created a binary classification task by regarding one class in a group as positive and the others in the same group as negative. We used multiple classes for negative data since negative data are often diverse in practice (Hsieh et al., 2019). By changing positive classes, we created 80 source, 20 validation, and 10 target tasks. We note that there is no data overlap between source, validation, and target tasks, and creating multiple tasks from the same dataset is a standard procedure in meta-learning studies (Kumagai et al., 2019; 2021; 2023; Finn et al., 2017; Snell et al., 2017). For Isolet, we first split all five domains into three, one, and one. Then, by using the same procedure for Mnist-r, we created 78 source, 26 validation, and 26 target tasks. Since there are significant task differences in each dataset (e.g., positive data in a task can be negative in other tasks), task-specific classifiers are required. For IoT, we directly split all nine domains into six source, two validation, and one target tasks.

For each source/validation task, the positive class-prior was uniformly randomly selected from $\{0.2, 0.4, 0.6, 0.8\}$. The number of data in each source/validation task of Synthetic, Mnist-r, Isolet, and IoT were 300, 120, 75, and 625, respectively. For each task, we treated half of the data as labeled data and the other half as unlabeled data. In each target task, we used 30 support instances (i.e., $N_{\mathcal{S}}^{\mathrm{p}} + N_{\mathcal{S}}^{\mathrm{u}} = 30$). We randomly created five different source/validation/target task splits for each dataset and evaluated mean test accuracy (1 - classification risk) on target tasks by changing the positive support set size $N_{\mathcal{S}}^{\mathrm{p}}$ within $\{1, 3, 5\}$ and the positive class-prior within $\{0.2, 0.4, 0.6, 0.8\}$ of target unlabeled/test data.

## 5.2 Comparison Methods

We compared the proposed method with the following nine methods: density-ratio-based PU learning method (DRE) (Charoenphakdee & Sugiyama, 2019), non-negative density-ratio-based PU learning method (DRPU) (Nakajima & Sugiyama, 2023), unbiased PU learning method (uPU) (Du Plessis et al., 2015), non-negative PU learning method (nnPU) (Kiryo et al., 2017), naive positive and negative learning method (Naive), three MAML-based PU leaning methods (MDRE, MDRPU, and MnnPU), and neural process-based PU learning method (NP). All comparison methods except for the proposed method use true positive class-prior information of target tasks.

DRE, DRPU, uPU, nnPU, and Naive use only target PU data for training. We evaluated these methods to investigate the effectiveness of using source data. If these methods outperform the proposed method, there is no need to perform meta-learning in the first place. Therefore, it is important to include these methods in comparison. DRE estimates the Bayes optimal classifier on the basis of a density-ratio estimation like the proposed method. To estimate the density-ratio, it minimizes the squared error between the true density-ratio and a RBF kernel model (Kanamori et al., 2009). DRPU is a density-ratio-based PU learning method with neural networks with a non-negative Bregman divergence estimator. Specifically, it minimizes the non-negative squared error, which is a modified squared error suitable for complex models such as neural networks. uPU is an unbiased PU learning method based on empirical risk minimization with the RBF kernel model. nnPU is a neural network-based PU learning method that minimizes the non-negative classification risk. Naive learns a neural network-based binary classifier by regarding unlabeled data as negative data. MDRE, MDRPU, and MnnPU are neural network-based meta-learning extensions of DRE, DRPU, and nnPU, respectively. Since there are no existing meta-learning methods for PU classification with a few PU data, we created these baseline methods. For the support set adaptation, MDRE and MDRPU perform density-ratio estimation with a gradient descent method. Unlike the proposed method, both methods adapt the whole neural networks for density-ratio estimation. MnnPU minimizes the loss function of nnPU with a gradient descent method. NP adapts to the support set by feed-forwarding it to neural networks like neural processes, which are widely used meta-learning methods (Garnelo et al., 2018). Specifically, NP uses $v([\mathbf{x}, \mathbf{z}^{\mathrm{p}}, \mathbf{z}^{\mathrm{u}}])$ as a classifier, where $v$ is a neural network and $\mathbf{z}^{\mathrm{p}}$ and $\mathbf{z}^{\mathrm{u}}$ are task representation vectors. All methods minimize the test classification risk in outer problems as in the proposed method. The details of neural network architectures and how to select hyperparameters are described in Sections B and C of the appendix.

## 5.3 Results

Table 1 shows average test accuracy over different target class-priors and positive support set sizes. The proposed method outperformed the other methods. Naive did not work well on all datasets, which indicates that simply treating unlabeled data as negative data introduces a strong bias. PU learning methods (DRE, DRPU, uPU, and nnPU) outperformed Naive, which indicates the effectiveness of PU learning. However, since they do not use source data, they performed worse than the proposed method. MDRE, MDRPU, MnnPU, and NP also utilize information from the source task as in the proposed methods. Since estimating binary classifiers from a few PU data is generally difficult, simply using meta-learning did not necessarily improve performance (e.g., see the results of MnnPU and nnPU). However, density-ratio-based meta-learning methods (the proposed method, MDRE, and MDRPU) outperformed their non-meta-learning variant (DRPU) with all datasets. Some existing studies have reported that density-ratio-based methods performed well when they used many data (Charoenphakdee & Sugiyama, 2019). Since meta-learning methods use many data in source tasks, the density-ratio-based meta-learning methods may perform well in our

Table 1: Average test accuracy [%] over different target class-priors within $\{0.2, 0.4, 0.6, 0.8\}$ and positive support set sizes within $\{1, 3, 5\}$. Boldface denotes the best and comparable methods according to the paired t-test ($p = 0.05$).

| Data | Ours | Naive | DRE | DRPU | uPU | nnPU | MDRE | MDRPU | MnnPU | NP |
|---|---|---|---|---|---|---|---|---|---|---|
| Synthetic | **82.37** | 66.05 | 73.84 | 73.66 | 76.98 | 78.97 | 79.82 | 78.96 | 77.90 | 80.24 |
| Mnist-r | **86.06** | 55.26 | 75.83 | 75.03 | 75.67 | 78.37 | **85.74** | **86.14** | 71.64 | 63.12 |
| Isolet | **93.08** | 54.92 | 85.80 | 82.35 | 84.00 | 84.73 | 91.29 | 92.45 | 78.33 | 75.19 |
| IoT | **98.70** | 66.05 | 76.52 | 74.10 | 71.37 | 76.74 | 96.40 | 97.54 | 97.26 | 98.38 |
| Average | **90.05** | 60.57 | 78.00 | 76.29 | 77.00 | 79.70 | 88.31 | 88.77 | 81.28 | 79.23 |

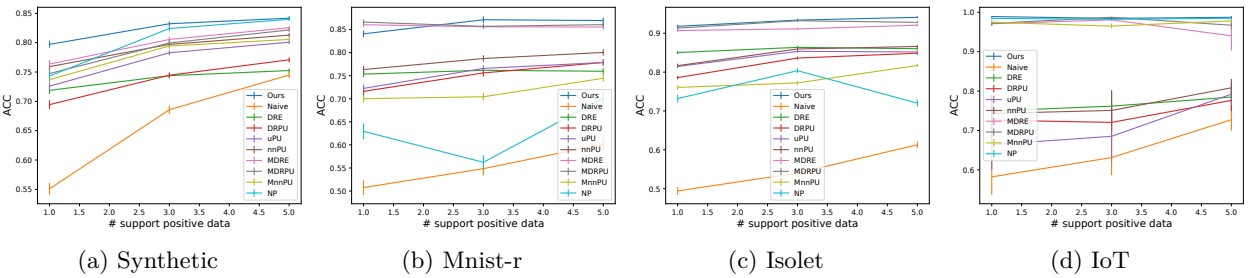

|(a) Synthetic|(b) Mnist-r|(c) Isolet|(d) IoT|

Figure 2: Average test accuracies and their standard errors when changing positive support set sizes.

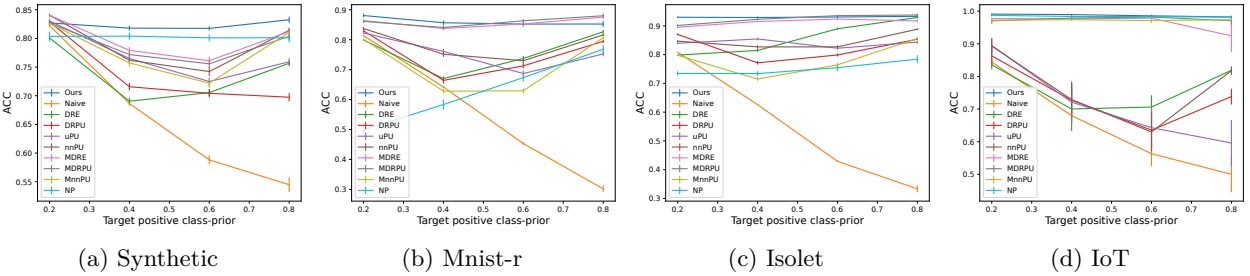

|(a) Synthetic|(b) Mnist-r|(c) Isolet|(d) IoT|

Figure 3: Average test accuracies and their standard errors when changing class-priors in target tasks.

experiments. These results suggest that density-ratio-based models are suitable for meta-learning of PU classification with a few PU data.

Figure 2 shows the average test accuracies and their standard errors when changing positive support set sizes $N_{\mathcal{S}}^{\mathrm{p}}$. All methods tended to improve performance as $N_{\mathcal{S}}^{\mathrm{p}}$ increased because labeled positive data are important for PU learning. The proposed method performed the best in almost all values of $N_{\mathcal{S}}^{\mathrm{p}}$. Note that although it seems that the proposed method and MDRPU yielded similar results due to the poor results of Naive in Isolet and IoT of Figure 2, the proposed was statistically better than MDRPU as in Table 1.

Figure 3 shows average test accuracies and their standard errors when changing positive class-priors on target tasks. Naive, which regards all unlabeled data as negative, greatly deteriorated performance as positive class-prior probability increased. This is because the number of positive data in the unlabeled data became large when the positive class-prior probability was large. The proposed method consistently performed well in almost all positive class-prior probabilities. The results of class-prior estimation performance of the proposed method are described in Section D.1 of the appendix.

Table 2 shows ablation studies of our model. We evaluated three variants of our model: w/o $\mathbf{z}$, w/o $\pi$, and w/ true $\pi$. w/o $\mathbf{z}$ is our model without task representations $\mathbf{z}^{\mathrm{p}}$ and $\mathbf{z}^{\mathrm{u}}$ in Eq. (4). w/o $\pi$ is our model without positive class-priors. This model uses $s_*(\mathbf{x}; \mathcal{S}) = \mathrm{sign}\left(\hat{r}_*(\mathbf{x}; \mathcal{S}) - 0.5\right)$ as a binary classifier. w/ true $\pi$ is our

Table 2: Ablation studies: Average test accuracy [%] over different target class-priors and positive support set sizes.

| Data | Ours | w/o $\mathbf{z}$ | w/o $\pi$ | w/ true $\pi$ |
|------|------|------|------|------|
| Synthetic | 82.37 | 75.92 | 79.01 | **84.78** |
| Mnist-r | **86.06** | 84.88 | 84.36 | **86.04** |
| Isolet | **93.08** | 92.04 | 91.51 | 91.86 |
| IoT | **98.70** | 98.21 | **98.73** | 98.36 |

Table 3: Training and testing time in seconds for meta-learning methods on Mnist-r

| | Ours | MDRE | MDRPU | MnnPU | NP |
|------|------|------|------|------|------|
| Train | 254.38 | 485.92 | 817.23 | 768.83 | 168.64 |
| Test | 0.018 | 0.047 | 0.053 | 0.050 | 0.012 |

model with true positive class-prior information. The proposed method outperformed w/o $\mathbf{z}$ and w/o $\pi$ on almost all datasets, which indicates the effectiveness of task representations and the estimated class-prior. Interestingly, the proposed method, which does not use true class-prior information, performed the same as or better than w/true $\pi$ in all datasets except for Synthetic. Since the class-prior estimation depends on the estimated density-ratio in Eq. (11), it might help to improve the performance of the density-ratio estimation. Since true class-priors are difficult or impossible to obtain without negative data in practice, this result shows the usefulness of the proposed method. The class-prior estimation performance of the proposed method was the worst in Synthetic: the average RMSEs of Synthetic, Mnist-r, Isolet, and IoT were 0.205, 0.179, 0.107, and 0.182, respectively. Thus, this might be one cause of the performance degradation in the ablation study. Since Synthetic has a large overlap in the positive and negative distributions, it may have been relatively difficult to estimate the class-prior distribution from a few PU data.

Table 3 shows the training and testing time in seconds of meta-learning methods with Mnist-r. We used a computer with a 2.20 GHz CPU. MAML-based methods (MDRE, MDRPU, and MnnPU) had much longer computation times than the others because they require costly gradient descent updates for adaptation. In contrast, since the proposed method can perform adaptation with the closed-form solution of the density-ratio estimation, it was more efficient than MDRE, MDRPU, and MnnPU. NP had the shortest computation time because it performs adaptation by simply feed-forwarding the support set to neural networks. However, the proposed method clearly outperformed NP in terms of accuracy by a large margin in Table 1.

## 6 Conclusion

In this paper, we proposed a meta-learning method for PU classification that can improve performance of binary classifiers obtained from PU data in unseen target tasks. The proposed method adapts to a few PU data by using only the closed-form solution of density-ratio estimation, leading to efficient and effective meta-learning. Experiments demonstrated that the proposed method outperforms various existing PU learning methods and their meta-learning extensions. As future work, we plan to extend our framework to treat different feature spaces across tasks by using techniques in the heterogeneus meta-learning (Iwata & Kumagai, 2020).

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

## A    Download Links of Real-world Datasets

We used three real-world datasets for our experiments: Mnist-r[2], Isolet[3], and IoT[4].

## B    Network Architecture

For the proposed method, a three(two)-layered feed-forward neural network was used for $f(g)$ in Eq. (3). For $f$, the number of hidden and output nodes was 100. For embedding network $h$ in Eq. (4), a four-layered feed-forward neural network with 100 hidden and output nodes was used ($J = 100$). The Softplus activation was used for the output nodes to ensure non-negativeness. For Naive, nnPU, MnnPU, and NP, a five-layered feed-forward neural network with 100 hidden nodes was used for classifier networks. For DRPU, MDRE, and MDRPU, a five-layered feed-forward neural network with 100 hidden nodes was used for modeling the density-ratio. For the task representation vectors in NP, the same neural network architectures as the proposed method ($f$ and $g$) were used. All neural networks used the ReLU function as their activation functions for the hidden nodes. We implemented all neural network-based methods on the basis of PyTorch (Paszke et al., 2017). All experiments were conducted on a Linux server with an Intel Xeon CPU and a NVIDIA A100 GPU.

## C    Hyperparameters

For meta-learning-based methods (the proposed method, MDRE, MDRPU, MnnPU, and NP), the hyperparameters were determined on the basis of mean validation accuracy. For non-meta-learning-based methods (Naive, DRE, DRPU, uPU, and nnPU), the best test results were reported from their hyperparameter candidates. We describe the hyperparameter candidates for each method below. For DRE and uPU, regularization parameter $\lambda$ was chosen from $\{10^{-3}, 10^{-2}, \ldots, 10\}$. Gaussian width of the RBF kernel was set to the median distance between support instances, which is a useful heuristic (median trick) (Schölkopf et al., 2002). For DRPU and MDRPU, non-negative correction parameter $\alpha$ was chosen from $\{0.1, 0.2, 0.3\}$. For Naive, DRPU, and nnPU, the number of training iterations was selected from $\{100, 500, 1000\}$. For the proposed method and NP, the dimension of task representations $K$ was chosen from $\{16, 32, 64, 128\}$. For MDRE, DRPU, and MnnPU, the iteration number for the support set adaptation was set to three, and the step size was selected from $\{10^{-2}, 10^{-1}, 1\}$. For density-ratio-based meta-learning methods (the proposed method, MDRE, and MDRPU), scaling parameter of the sigmoid function $\tau$ was set to 10. For all neural network-based methods, we used the Adam optimizer (Kingma & Ba, 2014) with a learning rate of $10^{-3}$. The mean validation accuracy was also used for early stopping to avoid over-fitting, where the maximum number of training iterations was $30,000$.

## D    Additional Experimental Results

### D.1    Class-prior Estimation Performance

Figure 4 shows average and standard errors of root mean squared errors (RMSEs) between true and estimated class-priors of the proposed method with different positive support set sizes. As $N_{\mathcal{S}}^{\mathrm{p}}$ increased, the proposed

---

[2]https://github.com/ghif/mtae
[3]http://archive.ics.uci.edu/ml/datasets/ISOLET
[4]https://archive.ics.uci.edu/ml/datasets/detection_of_IoT_botnet_attacks_N_BaIoT

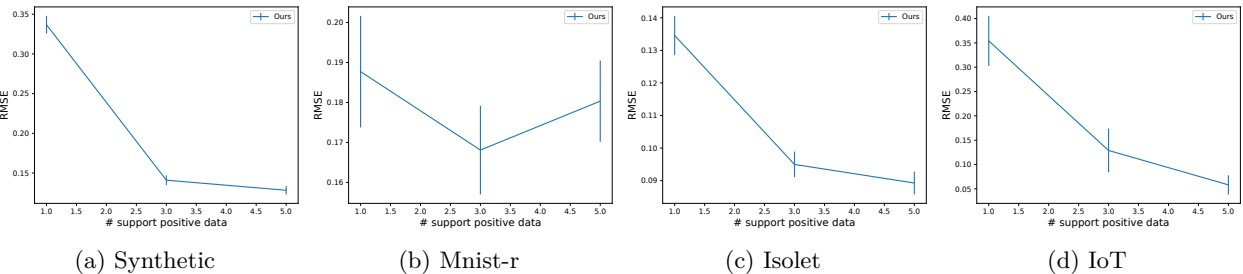

| (a) Synthetic | (b) Mnist-r | (c) Isolet | (d) IoT |

Figure 4: Average and standard errors of RMSEs between the true class-prior and estimated class-prior with the proposed method when changing positive support set size $N_{\mathcal{S}}^{\mathrm{p}}$.

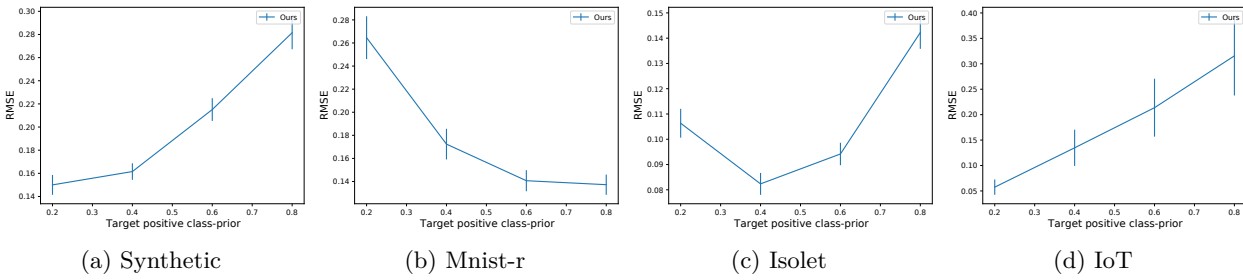

| (a) Synthetic | (b) Mnist-r | (c) Isolet | (d) IoT |

Figure 5: Average and standard errors of RMSEs between the true class-prior and estimated class-prior with the proposed method when changing positive class-priors in target tasks.

method tended to improve the estimation performance. This is because the density-ratio estimation used for class-prior estimation became more accurate as $N_{\mathcal{S}}^{\mathrm{p}}$ increased. Figure 5 shows average and standard errors of RMSEs between true and estimated class-priors with different positive class-priors on target tasks. Although the trend of RMSEs varied across datasets, in all cases, the RMSE was kept between 0.05 and 0.3 even with a few target PU data.

### D.2 Impact of Task Representation Vectors

Figure 6 shows average and standard errors of test accuracies when changing the dimension of task representation vectors $K$. The proposed method consistently performed better than DRE over all $K$ with all datasets. Figure 7 shows average and standard errors of RMSEs between truce and estimated class-prior of the proposed method when changing $K$. Although the best value of $K$ was varied across datasets, the difference in RMSE between different values of $K$ was small. These results demonstrate that the proposed method is relatively robust against $K$ values.

### D.3 Impact of Scaling Parameter $\tau$ in Sigmoid Function

The proposed method uses the smoothed test classification risk in Eq. (15) for meta-learning, which is obtained by replacing the zero-one loss in Eq. (14) with the sigmoid function with scaling parameter $\sigma_\tau(U) = \frac{1}{1+\exp(-\tau \cdot U)}$. We investigated the impact of $\tau$. Table 4 shows the results of the proposed method with $\tau = 1$ and $\tau = 10$. The proposed method with $\tau = 10$ outperformed it with $\tau = 1$. This is because our density-ratio-based score function $u_*(\mathbf{x}; \mathcal{S}, \Theta)$ used in the sigmoid function theoretically only takes values between $-0.5$ and $0.5$. In this case, the sigmoid function with small $\tau$ (e.g., 1), $\sigma_\tau$, cannot approximate the zero-one loss well (e.g., $\sigma_{\tau=1}(u_*(\mathbf{x}; \mathcal{S}, \Theta))$ takes values between 0.3775 and 0.6224). By using relatively large $\tau$ (e.g.,10), $\sigma_\tau$ can accurately approximate the zero-one loss (e.g., $\sigma_{\tau=10}(u_*(\mathbf{x}; \mathcal{S}, \Theta))$ takes values between 0.0067 and 0.9933). As a result, the proposed method with $\tau = 10$ performed well. We note that all

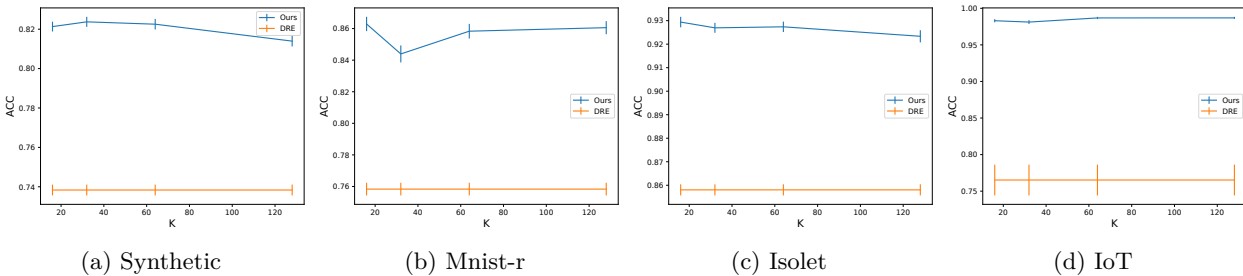

(a) Synthetic          (b) Mnist-r          (c) Isolet          (d) IoT

Figure 6: Average and standard errors of accuracies with the proposed method when changing the dimension of task representations $K$.

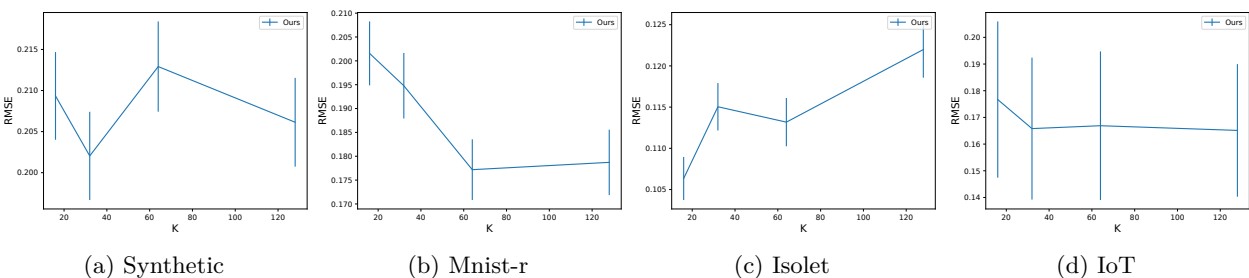

(a) Synthetic          (b) Mnist-r          (c) Isolet          (d) IoT

Figure 7: Average and standard errors of RMSEs between the true class-prior and estimated class-prior with the proposed method when changing the dimension of task representations $K$.

Table 4: The effect of value $\tau$ in the proposed method: Average test accuracy [%] over different class-priors and positive support set sizes. Boldface denotes the best and comparable methods according to the paired t-test and the significance level of 5 %.

| Data | Ours ($\tau = 10$) | Ours ($\tau = 1$) |
|---|---|---|
| Synthetic | **82.37** | 81.22 |
| Mnist-r | **86.06** | 84.00 |
| Isolet | **93.08** | 91.57 |
| IoT | **98.70** | **98.58** |

density-ratio-based meta-learning methods (the proposed method, MDRE, and MDRPU) used $\tau = 10$ in our experiments since they performed better than those with $\tau = 1$.

### D.4 Experiments with Small Label Ratios in Source Tasks

In the experiments of the main paper, we assume that half of the data in each source task is labeled. However, the number of labeled data might be smaller in practice. Table 5 shows average test accuracy when the labeled ratio was small (0.1) in each source task on IoT. We used IoT since the number of labeled data was insufficient to create positive support sets in the other real-world datasets. The proposed method achieved the best performance. This result shows that the proposed method works well even when the label ratio in source tasks is small.

### D.5 Experiments with a Larger Amount of Support Data

In the experiments in the main paper, we used a relatively small amount of support data. However, more support data might be available in practice. Table 6 shows average test accuracy with a larger support set on Synthetic. We used Synthetic since it has sufficient data in each task for preparing many support

Table 5: Results with small label ratio (= 0.1) in source tasks: Average test accuracy [%] over different class-priors in target tasks within $\{0.2, 0.4, 0.6, 0.8\}$ and positive support set sizes within $\{1, 3, 5\}$ on IoT. Boldface denotes the best and comparable methods according to the paired t-test and the significance level of 5 %.

| Ours | Naive | DRE | DRPU | uPU | nnPU | MDRE | MDRPU | MnnPU | NP |
|------|-------|-----|------|-----|------|------|-------|-------|-----|
| **96.59** | 66.05 | 76.52 | 74.10 | 71.37 | 76.74 | 95.14 | 95.41 | 95.88 | 96.23 |

Table 6: Results with a larger amount of support data: Average test accuracy [%] over target class-priors within $\{0.2, 0.4, 0.6, 0.8\}$ with different positive support set sizes $N_{\mathcal{S}}^{\mathrm{p}}$ within $\{10, 30, 50\}$ on Synthetic. We set $N_{\mathcal{S}}^{\mathrm{p}} + N_{\mathcal{S}}^{\mathrm{u}} = 100$. Boldface denotes the best and comparable methods according to the paired t-test and the significance level of 5 %.

| $N_{\mathcal{S}}^{\mathrm{p}}$ | Ours | Naive | DRE | DRPU | uPU | nnPU | MDRE | MDRPU | MnnPU | NP |
|------|------|-------|-----|------|-----|------|------|-------|-------|-----|
| 10 | **85.99** | 78.15 | 77.80 | 77.30 | 81.21 | 82.15 | 81.58 | 83.76 | 83.77 | 85.60 |
| 30 | **86.45** | 81.82 | 80.32 | 81.60 | 84.06 | 84.45 | 85.23 | 85.32 | 84.87 | 86.22 |
| 50 | **86.77** | 82.02 | 80.85 | 82.64 | 84.12 | 84.53 | 85.52 | 85.56 | 85.05 | 86.13 |

Table 7: Average test accuracies [%] with different numbers of source tasks $T$ in Synthetic.

| $T$ | Ours | Ours w/ true $\pi$ | nnPU |
|-----|------|--------------------|------|
| 100 | 82.37 | 84.78 | 78.97 |
| 75 | 81.72 | 84.61 | 78.97 |
| 50 | 81.45 | 83.86 | 78.97 |
| 25 | 77.13 | 80.01 | 78.97 |

(positive) data. The proposed method performed best even when support set sizes were large. Since positive support data is vital for PU learning, all methods tended to improve performances as $N_{\mathcal{S}}^{\mathrm{p}}$ increases. This result demonstrates the proposed method's effectiveness with relatively large support data.

## D.6  Experiments with Different Numbers of Source Tasks

Table 7 shows average test accuracies when changing the number of source tasks $T$ in Synthetic. We used Synthetic since it has sufficient source tasks ($T = 100$). As the number of source tasks increased, the proposed method (Ours) and it with true target class prior information (Ours w/o true $\pi$) performed better. This result indicates that the proposed method can improve its performance by gaining knowledge from many tasks. We note that Ours estimates the class priors of target tasks without knowing the true class priors. When $T$ was small (25), nnPU, which uses target PU data and true target class priors, slightly performed better than Ours. When using the true class priors, the proposed method (Ours w/ true $\pi$) outperformed nnPU. This result suggests the true class prior information is especially useful to construct accurate classifiers when the number of source data is not large. We note that the proposed method (Ours) worked better than nnPU in IoT, which has a small number of source tasks (6 tasks) in Table 1 in the main paper.

## D.7  Comparison with Task-invariant Classifiers

We compared the proposed method with a task-invariant approach, called Invariant, that trains a task-invariant classifier with all the source data. Since the meta-learning methods, including the proposed method, learn task-specific classifiers to handle task differences, this evaluation helps investigating the effectiveness of the meta-learning methods. Table 8 shows average test accuracy over different class-priors and positive support set sizes. The proposed method performed the best on all datasets. Invariant did not work well on all datasets except for IoT. This is because Synthetic, Mnist-r, and Isolet have significant task differences in each dataset (e.g., a positive class in one task can be negative in other tasks, etc.) as described in Section 5.1. Since IoT has small task differences, Invariant works well.

Table 8: Comparison with task-invariant classifiers: Average test accuracy [%] over different class-priors and positive support set sizes. Boldface denotes the best and comparable methods according to the paired t-test and the significance level of 5 %.

| Data | Ours | Invariant |
|------|------|-----------|
| Synthetic | **82.37** | 55.11 |
| Mnist-r | **86.06** | 50.01 |
| Isolet | **93.08** | 49.98 |
| IoT | **98.70** | **98.71** |

Table 9: Comparison with methods that use larger neural networks: Average test accuracy [%] over different class-priors and positive support set sizes. Boldface denotes the best and comparable methods according to the paired t-test and the significance level of 5 %.

| Data | Ours | MnnPU | MDRE | MDRPU |
|------|------|-------|------|-------|
| Synthetic | **82.37** | 73.12 | 79.47 | 79.79 |
| Mnist-r | **86.06** | 69.42 | 81.66 | 82.47 |
| Isolet | **93.08** | 73.31 | 83.53 | 85.22 |
| IoT | **98.70** | 92.11 | 98.03 | 97.13 |

### D.8 Comparison with Larger Neural Networks

In our experiments, for all neural network-based methods including the proposed method, we used five-layered feed-forward neural networks as a function from input $\mathbf{x}$ to output $y$ for a fair comparison. Here, $y$ is the density-ratio value for density-ratio-based methods and is the classification score for other methods. However, the proposed method and NP use additional neural networks $f$ and $g$ to compute the task representation $\mathbf{z}$. This difference in neural networks may affect the results. Therefore, we conducted a new experiment where the total number of layers of the neural network was matched for each method. Specifically, we used eight-layered feed-forward neural networks for meta-learning methods except for the proposed method and NP. Table 9 shows the average test accuracies with each dataset. The proposed method outperformed the others that used larger neural networks. These results provide further evidence of the effectiveness of the proposed method. We note that task-representation is not used for other methods because using itself is within our proposal.

## E Potential Applications of the Proposed Method

We discuss other potential real-world applications of the proposed method other than those described in Section 1.

We can consider recommendation systems for products to users. In this example, each user is regarded as a task. The user's past purchases are treated as positive data (of interest to the user), and unpurchased products are treated as unlabeled data. Since unpurchased products may contain products of interest to the user, they cannot be treated as negative data. The aim is to learn a user-specific classifier that predicts products of interest to the user that has a few positive data (purchased products). Some users may also provide information on products they are not interested in (i.e., negative data) as well as positive data. Such users can be used as source tasks.

We can consider medical diagnosis systems to predict if a patient has a disease in multiple hospitals. Since distributions of data among hospitals can differ due to the differences in patients or equipment (Chen et al., 2020b; Matsui et al., 2019), each hospital is treated as a task. The patients not diagnosed with the disease are treated as positive data, and others are treated as unlabeled data. Other patients cannot be treated as negative data because they may simply not have been tested. The aim is to learn a target hospital-specific classifier that predicts patients with the disease from a few PU data in a small hospital. Some hospitals may have patients diagnosed with the disease. These hospitals can be treated as source tasks.

In social networking services, we consider the problem of automatically discovering a user's friends. In this example, each user is regarded as a task. Although the users' friends (positive data) can be found using the user's friendship links, other users cannot be treated as non-friends (negative data) since they might contain some friends. The aim is to learn a user-specific classifier that predicts friends of the user that has little information about he/she friends. Some users may provide information about their non-friends by rejecting a recommendation of friends or blocking users. Such users can be treated as source tasks.

Although the proposed method assumes that the feature space is the same across tasks, it can be applied to many real-world applications described in Section 1 and above. For example, when data are images, the feature dimensions can be made the same by resizing. Thus, the proposed method can be applied to image retrieval, outlier detection for visual inspection, or medical image diagnosis. When data are text, we can treat the same feature (word) space across tasks within the same language. Thus, the proposed method can be applied to textual content-based recommendation systems or social networking services. In addition, even table data is often recorded in the same format (feature space) within a specific domain, such as a company or service. Thus, the proposed method would be applied.

## F    Limitations

The proposed method has a few limitations to overcome. First, the proposed method uses multiple source tasks to improve the classification performance on unseen target tasks. However, when target and source tasks are less related, the performance might degrade, which is a common limitation of existing meta-learning and transfer learning methods.

Second, the proposed method assumes that the feature space is the same across all tasks as in most existing meta-learning methods (Finn et al., 2017; Snell et al., 2017; Garnelo et al., 2018; Rajeswaran et al., 2019; Bertinetto et al., 2018; Kumagai et al., 2021). This assumption might be restrictive in practice. However, there are many real-world applications where the proposed method can be applicable, as described in Section 1. For example, when data are images, we can easily make the feature dimension of the task the same by resizing. When data are text, we can treat the same feature (word) space across tasks within the same language. As future work, we plan to develop a meta-learning method for PU classification that can treat tasks with different feature spaces as in (Iwata & Kumagai, 2020).

Third, the proposed method assumes that source tasks contain some negative data. We also plan to extend our framework such that it can handle source tasks that do not have negative data by rewriting the test classification risk with query PU data as in (Du Plessis et al., 2015; Kiryo et al., 2017; Du Plessis et al., 2014). Another idea is that when there is enough PU data for each source task, existing PU learning methods could be used as pre-processing to create pseudo-PN data on the source tasks for the proposed method.

## G    Social Negative Impacts

Although we demonstrated that the proposed method outperformed various PU learning methods in our experiments, it is not perfect; the proposed method has risks of misclassification. Thus, when the proposed method is used for mission critical applications, it should be used to support human decision making. In addition, since the proposed method uses data from multiple tasks, biased datasets risk being included, which might result in biased results. We encourage researchers to develop methods to automatically detect such biases.

