# OpenReview forum: "Meta-learning for Positive-unlabeled Classification"
_TMLR — Rejected by TMLR_

### Review · Reviewer_aQxf · 2024-02-11

**Summary Of Contributions:**

The paper proposes a meta learning method to estimate the density-ratio of PU data, and utilize the estimated density-ratio to obtain the class prior for PU classifier.

**Audience:**

Yes

**Broader Impact Concerns:**

There are no concerns on the ethical implications of this work.

**Claims And Evidence:**

Yes

**Requested Changes:**

1) The paper should give more real-world scenarios where the proposed method can be applied.
2) The paper should detail the density-ratio estimation process, why the linear combination of instances embeddings can be utilized to model the density-ratio.
3) The paper should perform more experiments, only four data sets are not enough. The data sets utilized in the comparison methods should be the same.

**Strengths And Weaknesses:**

Strengths:
1) The paper is well organized.

Weaknesses:
1) The proposed method seems not applicable in many scenarios, since it requires the source tasks and target tasks share the same distribution and the same feature space. However, in real-world, the distributions are more likely to be different in different tasks.
2) In the density-ratio estimation steps, the proposed method requires that the source tasks contain some negative data, which violates the rule of PU learning. Although, the paper implies that the source tasks and the target tasks are different, they still require that the target tasks should be similar enough to the source tasks.
3) In the experiments, the compared methods DRE, DRPU, uPU and nnPU only utilize target PU data for training, but the proposed method utilize the source tasks data, which is unfair and makes the results not persuasive enough.

---

> ### Author Response · Authors · 2024-03-13
> **Response 1**
>
> Thank you for your insightful and constructive feedback. We revised the paper according to the comments.
>
> >The proposed method seems not applicable in many scenarios, since it requires the source tasks and target tasks share the same distribution and the same feature space. However, in real-world, the distributions are more likely to be different in different tasks.
>
> Our method does not require that source and target tasks share the same distribution.
> As described in Section 4.1, our method assumes that the joint distribution of each task $p_t({\bf x},y)$ can vary.
> Although our method assumes the same feature space across tasks, this requirement is common in existing meta-learning methods (Finn et al., 2017; Snell et al., 2017; Garnelo et al., 2018; Rajeswaran et al., 2019; Kumagai et al., 2023; Farid & Majumdar, 2021). To clarify these, we revised a sentence in Section 4.1 as follows:
>
> "We assume that feature vector size $D$ is the same across all tasks, and all tasks are drawn from the same task distribution although joint distribution $p_t({\bf x},y)$ can vary across tasks.
> These assumptions are common in meta-learning studies (Finn et al., 2017; Snell et al., 2017; Garnelo et al., 2018; Rajeswaran et al., 2019; Kumagai et al., 2023; Farid & Majumdar, 2021)."
>
> In addition, even if the feature space is the same across all tasks, there are many possible applications where our method can be applied, as described in Sections E and F. For example, when data are images, we can easily make the feature dimension of the task the same by resizing.
> Thus, our method can be applied to image retrieval, outlier detection (for visual inspection), or medical image diagnosis, which are the motivating examples described in Sections 1 and E.
> Since our work is the first attempt at meta-learning for PU classification, we considered this setting (the same feature space across tasks) as the first step.
>
> To handle different feature spaces, we can consider incorporating the techniques of heterogeneous meta-learning (Iwata & Kumagai, 2020), which can treat different feature spaces across tasks, into our framework.
> We added the following sentence in Section 6 to clarify this.
>
> "As future work, we plan to extend our framework to treat different feature spaces across tasks by using techniques in the heterogeneous meta-learning (Iwata & Kumagai, 2020)."
>
> >In the density-ratio estimation steps, the proposed method requires that the source tasks contain some negative data, which violates the rule of PU learning.
>
> Our method does not require negative data in the density-ratio estimation step (Section 4.2.2).
> It requires only PU data in a task to obtain the Bayes optimal classifier. Although our method requires negative data in source tasks for meta-learning (Section 4.3), it requires only PU data in target tasks. Thus, our method does not violate the rule of PU learning in the target tasks. To clarify this, we added the following sentence in Section 4.2.3:
>
> "We note that estimating the Bayes optimal classifier, including the task representation calculation, the density-ratio estimation, and the class-prior estimation, requires only PU data. Therefore, our model can be applied to target tasks that have only PU data."
>
> In addition, when there is no negative data in source tasks, we can create pseudo-negative data from PU data by using the existing PU learning methods described in Section F.
> Thus, in this case, we might use these pseudo-negative data in our framework.
>
> >Although, the paper implies that the source tasks and the target tasks are different, they still require that the target tasks should be similar enough to the source tasks.
>
> Thank you for the insightful comment. As you commented, our method implicitly assumes that the source and target tasks are drawn from the same "task" distribution.
> This requirement is common in almost all meta-learning studies to ensure the generalization performance of meta-learning (Finn et al., 2017; Farid & Majumdar, 2021).
> To clarify this, as stated in the first answer in this response, we revised a sentence in Section 4.2.1 as follows:
>
> "We assume that feature vector size $D$ is the same across all tasks, and all tasks are drawn from the same task distribution although joint distribution $p_t({\bf x},y)$ can vary across tasks.
> These assumptions are common in meta-learning studies (Finn et al., 2017; Snell et al., 2017; Garnelo et al., 2018; Rajeswaran et al., 2019; Kumagai et al., 2023; Farid & Majumdar, 2021)."

---

> > ### Author Response · Authors · 2024-03-13
> > **Response 2**
> >
> > >In the experiments, the compared methods DRE, DRPU, uPU and nnPU only utilize target PU data for training, but the proposed method utilizes the source tasks data, which is unfair and makes the results not persuasive enough.
> >
> > > The data sets utilized in the comparison methods should be the same.
> >
> > As described in Section 5.2, we included these methods to investigate the effectiveness of using source data. If these methods outperform our method, there is no need to perform meta-learning in the first place. Therefore, it is important to include these methods in comparison for extensive experiments. To clarify this point, we added the following sentence in Section 5.2.
> >
> > "If these methods outperform the proposed method, there is no need to perform meta-learning in the first place. Therefore, it is important to include these methods in comparison."
> >
> > We note that MDRE, MDRPU, MnnPU, and NP in the comparison methods use both source and target data as in our method.
> >
> >
> > >The paper should give more real-world scenarios where the proposed method can be applied.
> >
> > We have discussed several real-world scenarios (outlier detection, information retrieval, recommendation system, medical diagnosis, and social networking services) where our method can be applied in Sections 1 and E.
> > We believe that these scenarios motivate our problem setting.
> > To state that the applications are discussed in the appendix (Section E), we added the following footnote in Section 1:
> >
> > "We discuss other potential applications of the proposed method in Section E of the appendix."
> >
> > In addition, to state that the assumption of our method (the feature space is the same across tasks) is still practical, we added the following sentence in Section E:
> >
> > "Although the proposed method assumes that the feature space is the same across tasks, it can be applied to many real-world applications described in Section 1 and above.
> > For example, when data are images, the feature dimensions can be made the same by resizing. Thus, the proposed method can be applied to image retrieval, outlier detection for visual inspection, or medical image diagnosis.
> > When data are text, we can treat the same feature (word) space across tasks within the same language. Thus, the proposed method can be applied to textual content-based recommendation systems or social networking services. In addition, even table data is often recorded in the same format (feature space) within a specific domain, such as a company or service. Thus, the proposed method would be applied."
> >
> > >The paper should detail the density-ratio estimation process, why the linear combination of instances embeddings can be utilized to model the density-ratio.
> >
> > First, in Eq. (4), we used a neural network for the density-ratio model so that complex density-ratios can be modeled. To clarify this, we added the following sentence in Section 4.2.2:
> >
> > "In particular, neural networks have been recently used for directly modeling the density-ratio due to their high flexibility and expressibility.
> > Following this, our method models the density-ratio by the following neural network".
> >
> > In addition, this density-ratio model in Eq. (4) consists of task-shared embedding network $h$ and task-specific liner weights ${\bf w}$.
> > By adapting linear weights ${\bf w}$ only to given support set ${\cal S}$, we can obtain the estimated density-ratio as a closed-form solution, which leads to efficient and effective meta-learning.
> > Therefore, we used this modeling.
> > To clarify this, we added the following sentences in Section 4.2.2:
> >
> > "Liner weights ${\bf w}$ and parameters of all neural networks $f$, $g$, and $h$ are task-specific and task-shared parameters, respectively. Task-shared parameters are meta-learned to improve the expected test performance, which is explained in Section 4.3. Only task-specific parameters ${\bf w}$ are adapted to given support set ${\cal S}$, which enables us to estimate the density-ratio as a closed-form solution."
> >
> > We used task representation vectors ${\bf z} ^p$ and ${\bf z} ^u$ to condition instance embedding $h$. Since ${\bf z} ^p$ and ${\bf z} ^u$ are calculated from a support set in a task, embedding $h(\cdot, {\bf z} ^p, {\bf z} ^u)$ can be task-specific, which is useful to deal with the tasks' diversity.
> > In fact, our method with the task representation vectors (Ours) outperformed our method without them (w/o ${\bf z}$) in our experiments (Table 2.)

---

> > > ### Author Response · Authors · 2024-03-13
> > > **Response 3**
> > >
> > > >The paper should perform more experiments, only four data sets are not enough.
> > >
> > > Thank you for the comment. Four datasets used in our experiments are sufficient as meta-learning studies because each dataset consists of many tasks (specifically, Synthetic, Mnist-r, Isolet, and IoT have 140, 110, 130, and 9 tasks, respectively).
> > > In fact, many meta-learning studies used about two or three real-world datasets (Kumagai et al., 2023; Iwata & Kumagai, 2020; Snell et al., 2017; Rajeswaran et al., 2019).
> > > Also, many recent PU learning studies used about three datasets (Chen et al., 2020b; Zhao et al., 2022; Hsieh et al., 2019; Wang et al, 2023).
> > > In addition, in this paper, many experiments were conducted in each dataset with various conditions, varying the number of support data (Figures 2 and 4, and Table 6), positive class priors (Figures 3 and 5), label ratios in source tasks (Table 5), and neural network sizes (Table 9). Thus, we think that our experiments are sufficient.

---

### Review · Reviewer_Qmyc · 2024-03-09

**Summary Of Contributions:**

The paper introduces a meta-learning approach for positive and unlabeled (PU) classification, aimed at improving binary classifiers' performance on unseen tasks with limited PU data. The key contribution is the development of a method that adapts models to new tasks by leveraging related tasks' data. The approach uses neural networks for task-specific instance embedding and density-ratio estimation, leading to efficient and effective model adaptation. The proposed method outperforms existing PU learning techniques

**Audience:**

Yes

**Claims And Evidence:**

Yes

**Requested Changes:**

- The manuscript requires a thorough revision of its notation system to ensure clarity and consistency, e.g., the current usage of superscripts and subscripts is ambiguous.

- Figure 1 needs significant enhancement. As it stands, the illustration does not adequately convey the operational flow or the key concepts underlying the algorithmic procedure.

**Strengths And Weaknesses:**

Strengths:

- The motivation is clearly articulated.

- This paper introduces an effective framework, supported by experimental results.

- The proposed algorithm benefits from a theoretical foundation, with each component of the framework analyzed.

Weaknesses:

- While the manuscript acknowledges the significance of established techniques within PU learning, such as density-ratio estimation and class-prior estimation, and appropriately cites related work, it falls short in clearly articulating the direct applicability of these methods to the meta PU problem.  It remains ambiguous whether these techniques are directly implemented or if modifications were necessary;  if the latter, the specific challenges and the novel contributions of the proposed solutions are not distinctly outlined.  This lack of clarity raises questions regarding the technical innovation of the proposed algorithm.

- Although the paper is rich in mathematical derivations to substantiate the algorithm's design, it lacks clear explanations for key equations, such as Eqs. 4 and 9.  The rationale for initiating the derivation process remains unclear, making the paper less accessible to non-expert readers.  This issue detracts from the overall readability and comprehensibility of the manuscript.

---

> ### Author Response · Authors · 2024-03-13
> **Response**
>
> Thank you for your thoughtful and constructive feedback. We revised the paper according to the comments.
>
> > While the manuscript acknowledges the significance of established techniques within PU learning, such as density-ratio estimation and class-prior estimation, and appropriately cites related work, it falls short in clearly articulating the direct applicability of these methods to the meta PU problem. It remains ambiguous whether these techniques are directly implemented or if modifications were necessary; if the latter, the specific challenges and the novel contributions of the proposed solutions are not distinctly outlined.
>
> The proposed method uses the established techniques of density-ratio estimation and class-prior estimation on a task-specific embedding space, which is inferred from a few PU data using neural networks, to obtain the Bayes optimal classifier.
> As long as the results of both density-ratio and class-prior estimations are differentiable w.r.t. the neural network parameters, our meta-learning framework can directly use those methods on the task-specific embedding space without modification.
> However, a naïve application (using MAML) would compromise computational cost (as shown in Table 3).
> The key point of the proposed method is that it achieves efficient and effective meta-learning by using the appropriate density-ratio estimation method that can estimate the density-ratio as a closed-form solution (Kanamori et al., 2009) and using the estimated density-ratio to estimate the class-prior.
> To clarify our contributions, we summarized our main contribution in Section 1 as follows:
>
> "Our main contributions are summarized as follows:
>
> - To the best of our knowledge, our work is the first attempt at meta-learning for positive-unlabeled classification with a few PU data.
> - We propose an efficient and effective meta-learning method that estimates the task-specific Bayes optimal classifier using only the closed-form solution of the density-ratio estimation.
> - We empirically show that the proposed method outperformed existing PU learning methods and their meta-learning variants when there is insufficient PU data in both synthetic and real-world datasets."
>
>
> >  Although the paper is rich in mathematical derivations to substantiate the algorithm's design, it lacks clear explanations for key equations, such as Eqs. 4 and 9. The rationale for initiating the derivation process remains unclear, making the paper less accessible to non-expert readers.
>
> Thank you for the constructive comment. We added the following sentences to explain the key equations (Eqs. (3,4,9)) well to improve readability.
>
> - For task representation vectors (Eq. 3)
>
> "First, we explain how to obtain a vector representation of the given dataset, which is used for obtaining task-specific instance embeddings appropriate for the task."
>
> - For density-ratio estimation (Eq. 4)
>
> "In particular, neural networks have been recently used for directly modeling the density-ratio due to their high flexibility and expressibility.
> Following this, our method models the density-ratio by the following neural network,"
>
> "Liner weights ${\bf w}$ and parameters of all neural networks $f$, $g$, and $h$ are task-specific and task-shared parameters, respectively. Task-shared parameters are meta-learned to improve the expected test performance, which is explained in Section 4.3. Only task-specific parameters ${\bf w}$ are adapted to given support set ${\cal S}$, which enables us to estimate the density-ratio as a closed-form solution."
>
> - For class-prior estimation (Eq. 9)
>
> "We explain how to estimate class-prior $\pi^{\rm p}$ from ${\cal S}$ by using estimated density-ratio ${\hat r} _{\ast} ({\bf x} ; {\cal S} )$.
> To view the relationship between the density-ratio and the class-prior, we first consider the following equation as in (Blanchard et al., 2010; Scott,
> 2015):"
>
> > The manuscript requires a thorough revision of its notation system to ensure clarity and consistency, e.g., the current usage of superscripts and subscripts is ambiguous.
>
> To enhance clarity and consistency, we revised the notations of the paper. Specifically, we consistently used superscripts for indexes about the positive, negative, and unlabeled distributions $({\rm p}, {\rm n}$, and ${\rm u})$ and subscripts for indexes about tasks and data ($t$ and $n$) in the revised paper.
> In addition, in the introduction of Section $4.2$, we described that task index $t$ is omitted for simplicity as follows.
>
> "We omit task index $t$ for simplicity since all procedures are conducted in a task in this section."
>
> > Figure 1 needs significant enhancement. As it stands, the illustration does not adequately convey the operational flow or the key concepts underlying the algorithmic procedure.
>
> Thank you for the constructive comment.
> We revised Figure 1 to clarify the key operations, such as density-ratio and class-prior estimations, for constructing the Bayes optimal classifier.

---

### Review · Reviewer_PweR · 2024-03-11

**Summary Of Contributions:**

This paper proposes a meta-learning method for PU learning. It targets the problem that sufficient data may not be available in real-practice PU learning and proposes to improve the performance from limited PU data by using multi-sourced related PNU data. The paper further analyzes the PU problem and decomposes it into density-ratio estimation and class-prior estimation. It proposes to use permutation-invariant neural networks to perform density-ratio estimation and class-prior estimation. With these estimations, the paper targets to get the Bayes optimal classifier. Experiments validate the effectiveness of the proposed method.

**Audience:**

Yes

**Claims And Evidence:**

Yes

**Requested Changes:**

1. The paper needs a more comprehensive discussion on its relationship with PU learning and PNU learning and cite/compare with the related methods.
2. More analysis of the experimental results may be helpful. For example, why MnnPU is worse than nnPU even when using meta-learning techniques (as shown in Table 1) is not sufficiently discussed.

**Strengths And Weaknesses:**

1. The paper decomposes the PU problem into density ratio estimation and class prior estimation to get the Bayes optimal classifier. Effective methods are proposed to estimate these values.
2. The paper proposes the insufficient data problem for PU learning, which is a novel problem not studied previously. It is also novel to use meta-learning to solve the problem.
3. The paper is well-organized and presented. The visualization is very clear as well, which is easy to follow.
4. The paper provided many experimental results to show the superiority of the proposed method from different perspectives.

Despite the strengths, the paper uses additional negative data, which may make the paper out of the classical PU learning scenario, but more in a PNU setting^1, where the most related paper is not cited or discussed/compared with.

[1] Semi-Supervised Classification Based on Classification from Positive and Unlabeled Data. ICML17

---

> ### Author Response · Authors · 2024-03-13
> **Response**
>
> Thank you for your positive and constructive feedback. We revised the paper according to the comments.
>
> >Despite the strengths, the paper uses additional negative data, which may make the paper out of the classical PU learning scenario, but more in a PNU setting, where the most related paper is not cited or discussed/compared with.
>
> >The paper needs a more comprehensive discussion on its relationship with PU learning and PNU learning and cite/compare with the related methods.
>
> Thank you for the constructive comment.
> The problem settings/purposes of our method and PNU learning methods are quite different:
> PNU learning (semi-supervised learning) assumes positive, negative, and unlabeled data in a target task. Also, it cannot use data in source tasks.
> On the other hand, our method assumes positive, negative, and unlabeled data in source tasks but only positive and unlabeled data for a target task.
> Since joint distribution $p_t(x,y)$ can differ across tasks in our setting, as described in Section 4.1, negative data in source tasks are not directly used as negative data in target tasks. Thus, PNU learning methods are inappropriate for our setting.
> We added the following discussion about the relationship between our method and PNU learning methods in Section 2.
>
> "PNU learning uses positive, negative, and unlabeled (PNU) data in a task to learn binary classifiers (Sakai et al., 2017; Hsieh et al., 2019; Sakai et al., 2018). Especially, (Sakai et al., 2017) uses the technique of PU learning to rewrite the classification risk with PNU data, which enables us to learn binary classifiers without particular distributional assumptions such as the cluster assumption (Grandvalet & Bengio, 2004). Unlike the proposed method, these methods cannot use data in source tasks and require negative data in the target tasks."
>
> > More analysis of the experimental results may be helpful. For example, why MnnPU is worse than nnPU even when using meta-learning techniques (as shown in Table 1) is not sufficiently discussed.
>
> We added some discussion about our experimental results in Section 6 as follows:
>
> "Since estimating binary classifiers from a few PU data is generally difficult, simply using meta-learning did not necessarily improve performance (e.g., see the results of MnnPU and nnPU). However, density-ratio-based meta-learning methods (the proposed method, MDRE, and MDRPU) outperformed their non-meta-learning variant (DRPU) with all datasets.
> Some existing studies have reported that density-ratio-based methods performed well when they used many data (Charoenphakdee & Sugiyama, 2019). Since meta-learning methods use many data in source tasks, the density-ratio-based meta-learning methods may perform well in our experiments.
> These results suggest that density-ratio-based models are suitable for meta-learning of PU classification with a few PU data."

---

### Decision · Action_Editor_6s1X · 2024-05-27

**Recommendation:** Reject

**Comment:**

The paper proposes a novel meta-learning method to improve binary classifiers' performance when applied to only positive and unlabeled (PU) data on unseen target tasks, using positive, negative and unlabeled (PNU) data during meta-training. The core thought is to only use the linear layer for adaption, allowing an efficient estimate of the density ratio. The density ratio is also used to estimate the class prior, making an efficient total solution for the adaptation task. Meta-training part appears standard, except for plugging in the adaptation routines during episode training. The authors obtained stronger results for this PNU->PU meta-learning scenario when compared with baselines that are mostly non-meta-learning or PU->PU meta-learning.

The reviewers and the action editor agree on the following strength:
* The authors present an interesting novel problem of PNU->PU that can be potentially useful.
* The experiment results appear strong.

The reviewers and the action editor have concerns on the following issues:

* The PNU->PU setup is not introduced and differentiated with PU->PU or PNU->PNU clearly, causing significant reviewer confusion on whether the empirical results are fair or not. Actually, the authors should consider changing the title to "Meta-learning from Positive-Negative-Unlabeled Data for Positive-Unlabeled Classification" as the original title can be easily confused as working on the PU->PU setup.

* There is arguably not enough literature survey to understand *whether the paper belongs to combining existing solutions* or *designing a new solution*. It is fine if the paper belongs to the former (given TMLR's criteria), but the authors should state so clearly instead of reinventing the wheels. For instance, linear density ratio estimation seems to be long-standing in the literature (and so is the least-squared close-form solution), but Section 4.2.2 makes it look as if the authors invented the close-form solution by themselves. There is also not enough literature survey on how the authors choose competitors for Section 5.2---what their key differences and similarities to the proposed solution are.

* For the PNU part, the related literature is not discussed as well. The authors should also justify the difficulty of moving from PNU->PNU to PNU->PU.

* Overall the paper fails to provide evidence to *why the problem cannot be directly solved by existing meta-learning techniques* and how the solutions designed/chosen by the authors filled the gap. Some reviewers also believe that the technical clarity in terms of expressing the key insights behind the math can be improved.

* The reviewers are suspicious on the applicability in the real-world, despite the authors' efforts on simulating from three real-world datasets. More experiments on some more diverse real-world problems can strengthen the claim, as three datasets with thousands of instances in each dataset is insufficient in today's standard. While the authors cite a few papers that are working on the scale of about three datasets, many of them are on more challenging datasets (e.g. CIFAR-10) than what the authors are using (MNIST). So moving to some more challenging datasets is needed.

* More careful "ablation" studies on why the proposed solution outperforms MDRE (or MDRPU) seems needed. In particular, there are several differences between the proposed solution and MDRE, including whether negative data is used during meta-training, whether to adapt the linear layer only, whether to estimate the ratio with a RBF network model, etc. There is very little evidence on which difference caused the performance gap.

Overall, the work seems to have some potential, but requires a significant revision before acceptance.

**Audience:**

Yes

**Claims And Evidence:**

No, more experiments are needed to justify the applicability of the proposed solution, and to diffrentiate it with existing alternatives.

**Resubmission Of Major Revision:**

The authors may consider submitting a major revision at a later time.